# Synchronization in renal microcirculation unveiled with high-resolution blood flow imaging

**Dmitry Postnov[1,2]\*, Donald J Marsh[3], Will A Cupples[4], Niels-Henrik Holstein-Rathlou[2], Olga Sosnovtseva[2]**

[1]Department of Clinical Medicine, Faculty of Health, Aarhus University, Aarhus, Denmark; [2]Department of Biomedical Sciences, Faculty of Health and Medical Sciences, Copenhagen University, Copenhagen, Denmark; [3]Division of Biology and Medicine, Brown University, Providence, United States; [4]Department of Biomedical Physiology and Kinesiology, Simon Fraser University, Burnaby, Canada

**Abstract** Internephron interaction is fundamental for kidney function. Earlier studies have shown that nephrons signal to each other, synchronize over short distances, and potentially form large synchronized clusters. Such clusters would play an important role in renal autoregulation, but due to the technological limitations, their presence is yet to be confirmed. In the present study, we introduce an approach for high-resolution laser speckle imaging of renal blood flow and apply it to estimate the frequency and phase differences in rat kidney microcirculation under different conditions. The analysis unveiled the spatial and temporal evolution of synchronized blood flow clusters of various sizes, including the formation of large (>90 vessels) and long-lived clusters (>10 periods) locked at the frequency of the tubular glomerular feedback mechanism. Administration of vasoactive agents caused significant changes in the synchronization patterns and, thus, in nephrons' co-operative dynamics. Specifically, infusion of vasoconstrictor angiotensin II promoted stronger synchronization, while acetylcholine caused complete desynchronization. The results confirm the presence of the local synchronization in the renal microcirculatory blood flow and that it changes depending on the condition of the vascular network and the blood pressure, which will have further implications for the role of such synchronization in pathologies development.

**\*For correspondence:** dpostnov@cfin.au.dk

**Competing interest:** The authors declare that no competing interests exist.

## Editor's evaluation

The new technical advance reported here, high-resolution laser speckle contrast imaging of many microvessels simultaneously and application to the entire rat kidney surface should be of great interest to readers interested in general vascular physiology and especially renal hemodynamics research. The observed large vessel cluster synchronization and phase wave formation phenomena will help further the understanding of renal blood flow regulation in health and disease.

## Introduction

The kidney represents a unique demand-driven, interconnected resource distribution network that is responsible for body homeostasis maintenance over a broad range of conditions, including variations in blood pressure and fluid intake and loss. With blood serving as the resource, single-nephron autoregulation mechanisms provide and regulate the demand. Based on measurements of tubule pressure responses to step changes in arterial pressure (*Young and Marsh, 1981*), vascular transfer functions (*Chon et al., 1993*; *Chon et al., 2005*; *Cupples and Loutzenhiser, 1998*; *Marmarelis*

*et al., 1999*; *Sakai et al., 1986*; *Shi et al., 2006*), and renal blood flow response to arterial pressure forcing (*Holstein-Rathlou et al., 1991*), two critical mechanisms in renal pressure autoregulation, the myogenic mechanism and tubuloglomerular feedback (TGF), have emerged as the critical components. Their actions combine to regulate blood flow, serving to maintain the delivery of water and solutes to various regions of the nephron at levels appropriate to their dynamic ranges. The two mechanisms operate at different time scales, generating spontaneous blood flow and pressure oscillations at different frequencies: 5–10 s (0.1–0.2 Hz) for the myogenic response and 30–50 s (0.02–0.033 Hz) for the TGF (*Holstein-Rathlou and Marsh, 1989*; *Yip et al., 1993*; *Cupples and Loutzenhiser, 1998*; *Sosnovtseva et al., 2002*; *Just, 2007*).

Nephrons, however, do not operate as stand-alone units. Within a single kidney, all nephrons are linked via the renal vascular tree, which provides connections of different proximities - from few hundred microns for neighboring nephrons separated only by their respective afferent arterioles to nephrons only connected at the level of the renal artery, which plays a role of a single-supply source for all the nephrons. Nephrons nested in such a network are bound to communicate and affect each other to some degree. In addition to interaction through the blood flow and pressure, nephrons were found to communicate via electrical signaling (*Marsh et al., 2019*; *Marsh et al., 2009*). Such interactions are proven to play a critical role in kidney function and can lead to complex co-operative dynamics and synchronization between nephrons. Adaptive synchronization across the renal microvascular network would increase the efficiency and dynamic range of autoregulation by engaging more preglomerular resistance (*Marsh et al., 2019*; *Zehra et al., 2021*), preventing transmission of high systemic pressure to the glomeruli, which could lead to progressive glomerular and vascular injury. In addition, long-distance synchronization would argue strongly that renal autoregulation is a distributed process that can ensure an optimized oxygenation-perfusion matching and adjust to various internal or environmental conditions (*Sosnovtseva et al., 2007*; *Mitrou et al., 2015*). On the other hand, large-scale in-phase synchronization might become a mechanism leading to glomerular injury as it would substantially increase local pressure variation (*Postnov et al., 2012*).

Micropuncture experiments (*Holstein-Rathlou, 1987*; *Källskog and Marsh, 1990*; *Yip et al., 1992*; *Sosnovtseva et al., 2007*) showed that neighboring nephrons (originating from a common artery) adjust their TGF-mediated tubular pressure oscillations to attain a synchronized regime. Although these experiments confirmed the existence of synchronization, only pairs or triplets of nephrons could be sampled at any one time. Assessment of co-operative efforts of a larger number of nephrons required a different approach and was made possible with laser speckle contrast imaging (LSCI) technology (*Holstein-Rathlou et al., 2011*; *Mitrou et al., 2015*; *Scully et al., 2013*). In LSCI, media with moving light scattering particles, e.g., red blood cells, are illuminated with a near-infrared laser. The backscattered light, recorded by a camera, forms an interference pattern, which appears more or less blurred depending on the speed of the particles. This pattern is then analysed to obtain qualitative maps of particle velocity and thus a blood flow estimate (*Boas and Dunn, 2010*; *Briers et al., 2013*). First applied by Holstein-Rathlou et al. to map TGF oscillations over a large field of view, this method was later used to explore periodic activity in the myogenic frequency band (*Scully et al., 2013*; *Mitrou et al., 2015*), analyse spatial correlations in the renal blood flow (*Brazhe et al., 2014*), and study intrarenal drug distribution (*Postnov et al., 2015a*, *Postnov et al., 2017*). Although these results encouraged the large-scale synchronization hypothesis, the method lacked resolution, both spatial and temporal, as well as signal-to-noise ratio, to confirm it convincingly.

In this paper, we further advance renal blood flow imaging methodology and confirm the presence of synchronized clusters spanning multiple nephrons for the first time at the level of individual arterioles and venules. Our LSCI setup and data processing approach allow imaging renal microcirculation at 0.8 μm per pixel spatial resolution and imaging frequency to 160 Hz for 1024 × 1024 pixels. We apply it to study the formation of synchronous blood flow clusters in rat kidney microcirculation under normal conditions and during the administration of vasoactive agents.

## Results

### Synchronization patterns

An example of a high-resolution blood flow map with segmented microcirculatory vessels is shown in *Figure 1A*. *Figure 1B* shows relative blood flow (RBF) dynamics of four vessels outlined in (A) with

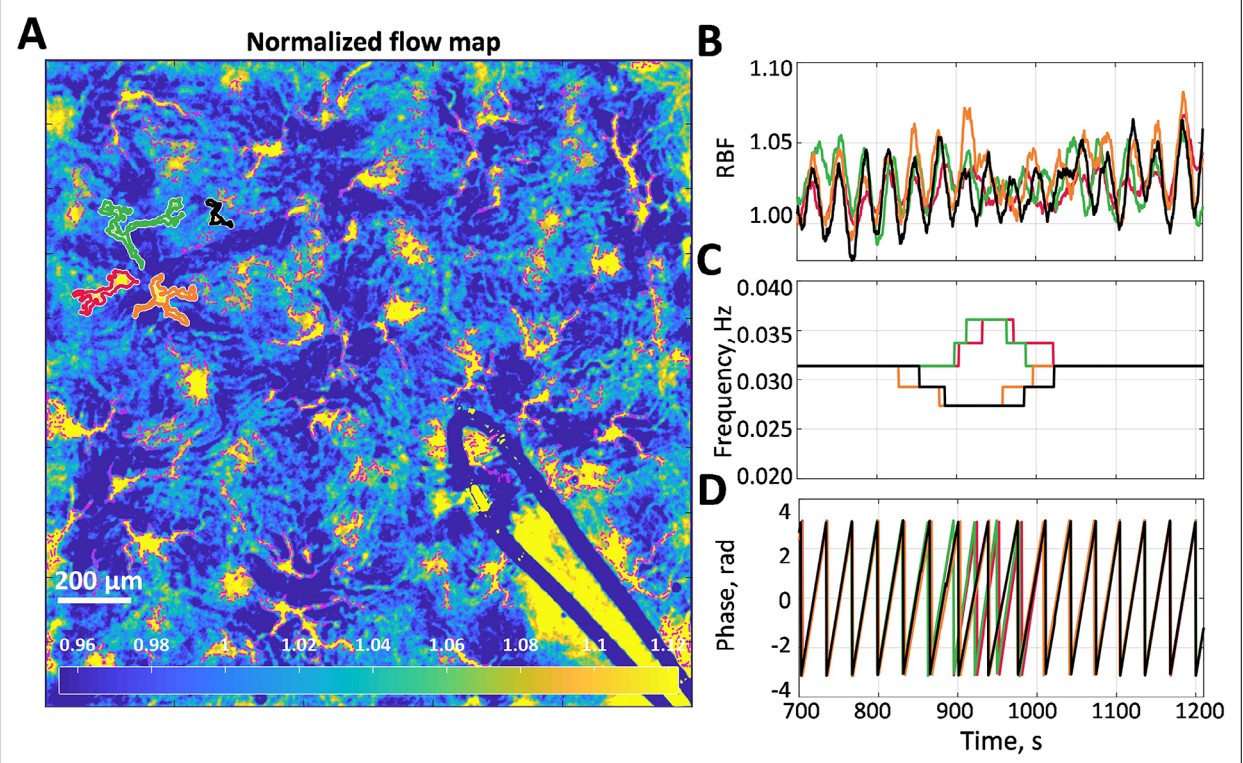

**Figure 1.** Blood flow data example. (**A**) Normalized flow map with microcirculatory vessels. Yellow and blue colors correspond to high and low blood flow, respectively. (**B**) Relative blood flow (RBF) dynamics in four vessels, which are outlined in (A) with corresponding colors. (**C**) Dominant frequency in the tubuloglomerular feedback (TGF) band of the blood flow oscillations is shown in (B). (**D**) Phase value at the dominant frequency. One can see that there are two vessel pairs in which blood flow is synchronized for more than 90% of the time. Synchronization between the pairs is also well observed but breaks down when TGF activity change its frequency at 850–1050 s.

The online version of this article includes the following figure supplement(s) for figure 1:

**Figure supplement 1.** Example of the data collected with the same imaging system (FLPI 1, Moor Instruments Ltd, UK) as used in the previous studies (*Holstein-Rathlou et al., 2011*; *Scully et al., 2013*; *Mitrou et al., 2015*) versus the data shown in the present study.

**Figure supplement 2.** Average changes in blood flow index.

corresponding colors (red, green, orange, and black). From the corresponding dominant frequency and phase (*Figure 1C and F*), it can be seen that these vessels form two frequency-locked clusters (red-green and black-orange) that are synchronous most of the time. Moreover, there are long periods when these clusters synchronize, forming a larger cluster, interrupted with an asynchronous interval (from ≈850 to ≈1050 s).

We generalized this approach to the full-field blood flow imaging of the kidney surface. To simplify the visualisation of frequencies and phases, we generate maps where colored circles represent the segmented vessels (outlined in magenta in *Figure 1A*), with the circle center located in the center of the mass of the vessel. The circle's color represents the peak TGF frequency observed in the corresponding vessel, while the size of the circles is identical and does not hold any information. *Figure 2* shows examples of high (A), moderate (B), and low (C) instantaneous synchronization degree (see Methods, Eq. 1), with S = 0.97, 0.65, and 0.23, respectively. In the first case, flow in almost all of the identified vessels oscillates at the same TGF frequency, forming a cluster of >100 vessels within the field of view. This cluster is likely to be even larger since it can spread outside the view field and in-depth in the kidney cortex. In the case of moderate synchronization degree, several frequency clusters with ≈ 10–60 vessels can be identified, while for the low S, there are multiple groups of two to three vessels displaying the same frequency but no distinct pattern over the field of view. All three regimes were observed in the same animal during control, vasoconstricted (angiotensin II [AngII] infusion), and vaso-dilated (acetylcholine [ACh] infusion) conditions, respectively. Averaged over the whole observation period and all animals (N = 5, 20 min per condition), synchronization degree has moderate to high

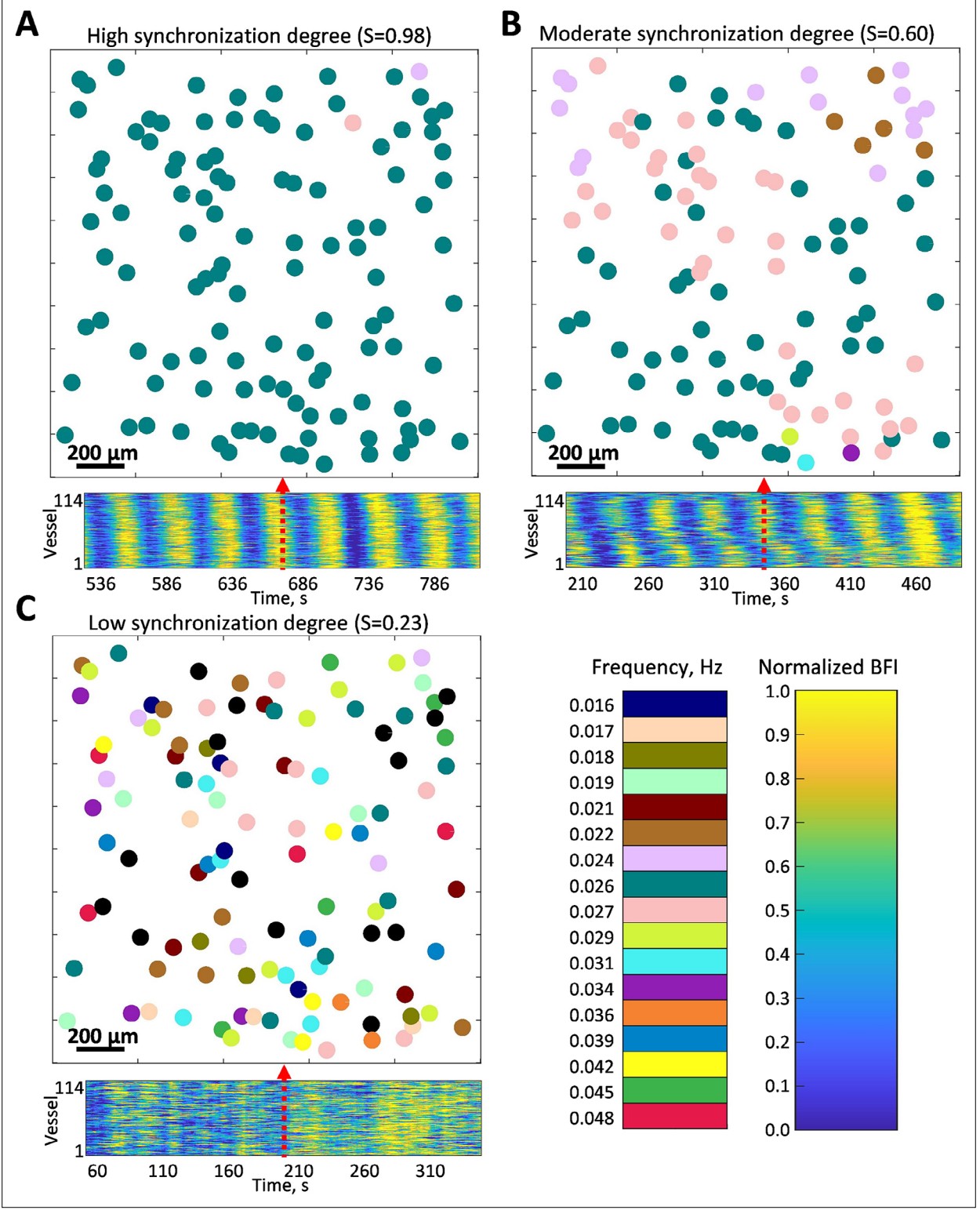

**Figure 2.** Examples of instantaneous frequency patterns. Each circle location corresponds to a segmented vessel with color-coded frequency. Bottom panels show normalized blood flow in each segmented vessel over 5 min centered around the time corresponding to the frequency pattern snapshot. (**A**) High synchronization degree $S = 0.97$ - large cluster covers the whole field of view. (**B**) Moderate synchronization degree $S = 0.65$ - most of the vessels are split between two clusters. (**C**) Low synchronization degree $S = 0.23$ - no clear clustering pattern, some vessels do not have pronounced tubuloglomerular feedback activity (frequency color-coded as black). All data were captured in the same animal but under different conditions: control (**A**), angiotensin II infusion (**B**), and acetylcholine infusion (**C**). Black-colored circles represent vessels where TGF activity was considered too weak (less than 10% prominence of the activity peak).

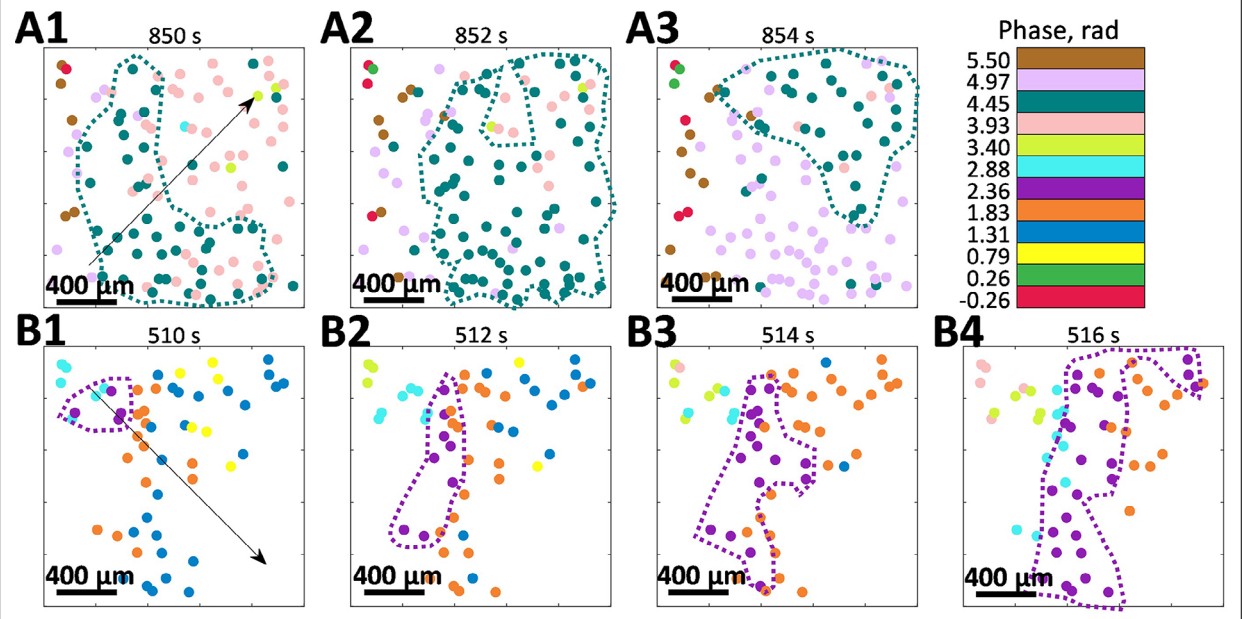

**Figure 3.** Probability of a randomly chosen vessel displaying the same dominant tubuloglomerular feedback (TGF) frequency as X% of the vessels in the field of view. (**A**) Any duration of frequency matching is considered, and (**B**) dominant frequency should match for at least three TGF periods. It can be seen that infusion of angiotensin II significantly (p<0.05) increases the prevalence of clusters covering 20–50% of the vessels - chances that a randomly chosen vessel belongs to such a cluster at any given moment are ≈50%, while for control and acetylcholine infusion they are ≈20%. N = 5 animals were used to create these graphs. Paired t-test was used to produce p-values. p-values smaller than 0.05 are considered to be significant and marked with '*'.

($\overline{S} = 0.54 \pm 0.09$) values during AngII infusion, low to moderate ($\overline{S} = 0.36 \pm 0.1$) during ACh, and low to high in control ($\overline{S} = 0.45 \pm 0.22$). The average TGF frequencies were measured to be $0.0271 \pm 0.0059$ Hz in control, $0.0270 \pm 0.0042$ during the AngII infusion, and $0.0256 \pm 0.0079$ during the ACh infusion.

To visualize clustering tendencies, we calculate the probability of a randomly chosen vessel at any given moment of time to have the same dominant frequency as X% of the vessels in the field of view. From *Figure 3*, it can be seen that during AngII infusion, vessels are significantly more likely to be synchronized with 20–50% of the field of view than in control or during ACh infusion. In the latter condition, probability of a vessel being synchronized with less than 15% is significantly higher than during AngII infusion (p<0.05). Such behavior is observed both with no restriction on minimum synchronization duration (A) and when only frequency-locking for three TGF periods and longer is taken into account.

## Phase waves and spatial localization

Another distinct feature that we have observed is that the phase of oscillations within a cluster is space-dependent - it is mostly the same for the closely positioned vessels and gradually changes with distance. *Figure 4* illustrates how phase within clusters evolves in space and time, forming the phase waves. Note that both (A) and (B) panels are from the same animal as was shown in *Figure 3* in control and vasoconstricted conditions, respectively, and that only vessels belonging to the largest cluster are shown.

Change of the wave direction by ≈90° in the same animal with constant vascular structure suggests different synchronization centers, where phase waves originate. Propagation speed is ≈ 0.37 and 0.30 mm/s for (A) and (B), respectively, and spatial period of the phase waves is ≈ 8.85 and 3.95 mm (respective phase difference |δ| ≈ 0.71 and 1.59 rad was observed over 1 mm distance). The possibility of such phase waves was previously hinted at in our earlier LSCI study, where spatial correlations in renal blood flow were analyzed (*Brazhe et al., 2014*). See *Figure 4—videos 1–3* for more examples of phase waves dynamics.

Phase difference distribution (*Figure 5A*), calculated over all of the frequency matching vessels in all animals, shows the prevalence of in-phase synchronization ($|\Delta| \leq \frac{\pi}{12}$, rad). This result is in good agreement with experimental micropuncture observations, where it is explained by the presence of

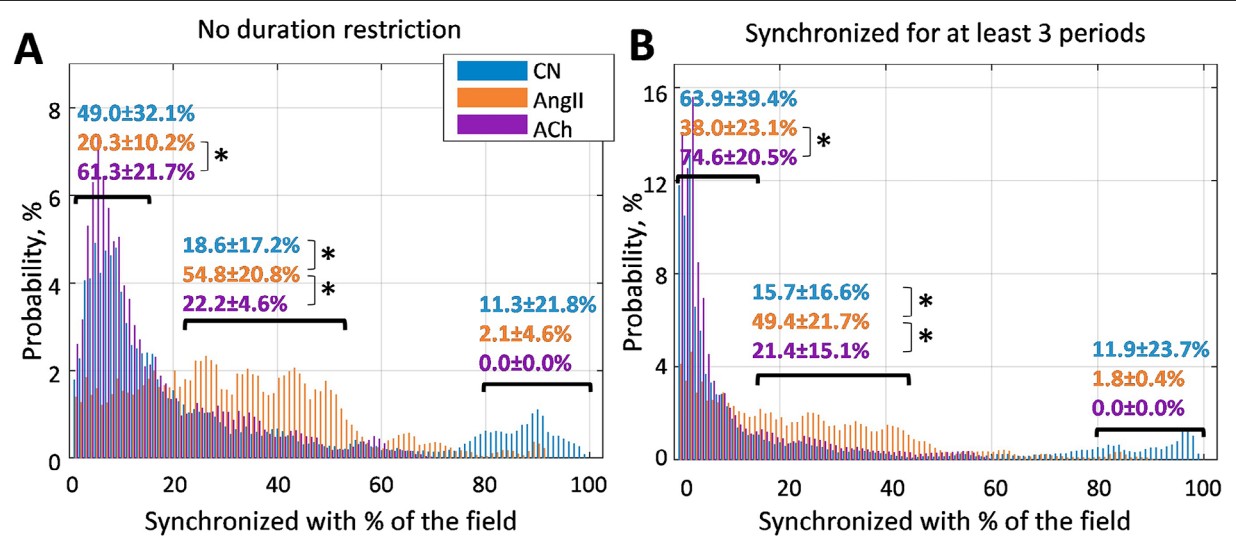

**Figure 4.** Phase waves. (**A**) and (**B**) show the spatio-temporal evolution of tubuloglomerular feedback activity phase in vessels that belong to the largest cluster observed during control and angiotensin II infusion. Both observations are from the same animal as shown in *Figure 2A, B*. Arrows indicate the wave direction.

The online version of this article includes the following video for figure 4:

**Figure 4—video 1.** Example of phase dynamics and phase waves visualization during angiotensin II (AngII) infusion.

https://elifesciences.org/articles/75284/figures#fig4video1

**Figure 4—video 2.** Example of phase dynamics and phase waves visualization in control.

https://elifesciences.org/articles/75284/figures#fig4video2

**Figure 4—video 3.** Example of phase dynamics during acetylcholine (ACh) infusion.

https://elifesciences.org/articles/75284/figures#fig4video3

fast electrical coupling acting over short distances. In all conditions, the larger phase differences are less prevalent, with the anti-phase synchronization ($|\Delta| \geq \frac{11\pi}{12}$, rad) observed over just 1% of the time in control and during ACh infusion. However, AngII infusion increases this number to 5%, showing a statistically significant difference with other conditions. Phase difference grows with distance, as can be seen from *Figure 5B*, reflecting the presence of phase waves. While change is relatively small in the control and vasodilated conditions (≈0.3 and 0.25 rad/mm), it is strongly enhanced in the vasoconstricted condition, reaching, on average, ≈1 rad over 1 mm distance. As we showed with mathematical modeling (*Postnov et al., 2012*), such difference can be explained by strengthened hemodynamic coupling, which the increased vascular tone should cause.

Since synchronization in renal blood flow is extended far beyond a pair of nephrons and is unlike synchronization in relatively homogeneous media, one would expect some space localization due to the topological features of the vascular network. *Figure 5C* illustrates how average synchronization duration changes with distance. It is clear that synchronization is stronger in the vasoconstricted condition, with average duration reaching ≈40±13% of the observation time for neighboring vessels and gradually reducing to 24±6% for vessels located at more than 1 mm distance. In control, synchronization duration varies greatly, but on average, it also reduces with distance, although at a slower rate than during AngII infusion—from ≈30% to ≈21%. During the ACh infusion the synchronization duration is ≈15-13%, with only 2% reduction over 1mm of distance. Higher synchronization duration in control and during AngII infusion compared with ACh infusion suggests a long-distance nephron-to-nephron communication or a common driving force.

## Discussion

In this study, we have designed a methodology for high-resolution blood flow imaging in renal microcirculation and applied it to study the synchronization of TGF oscillations in control, vasoconstricted

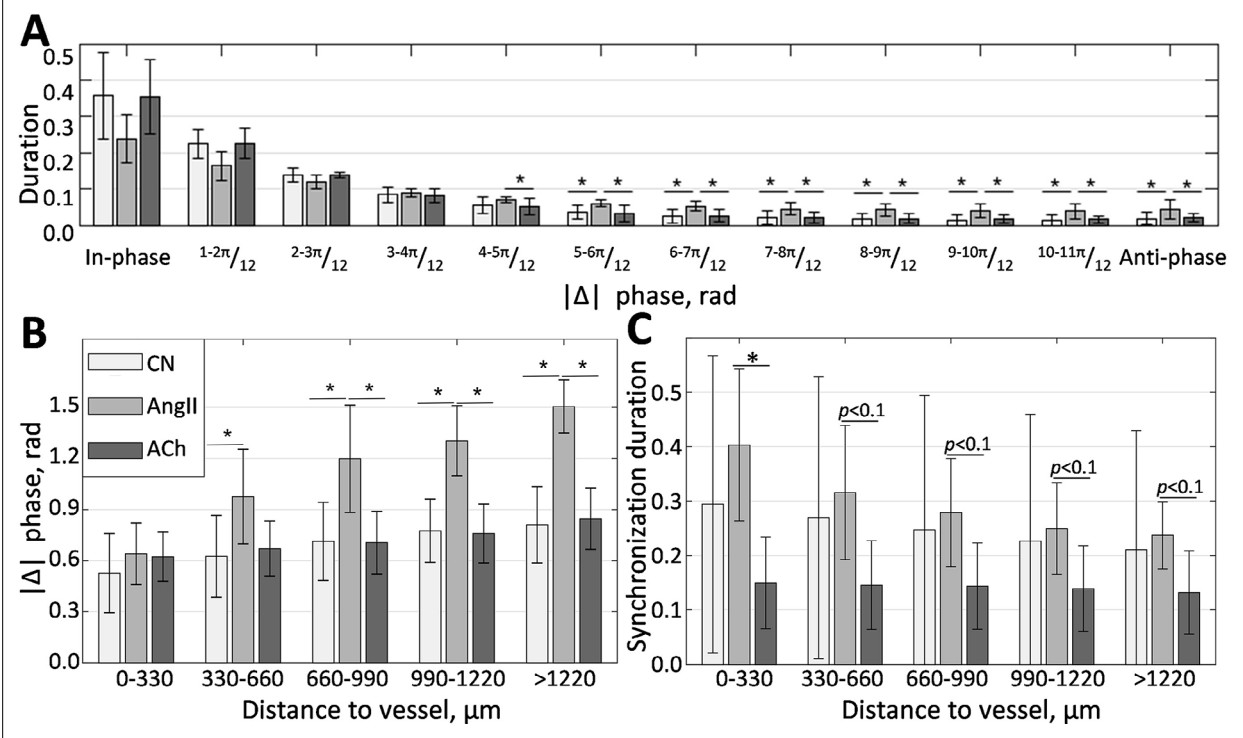

**Figure 5.** Localization of phase and synchronization in space. (**A**) Phase differences prevalence for synchronized vessels. (**B**) Phase differences distribution over the distance between vessels. (**C**) Synchronization duration normalized by the total observation time for different distances between vessels. N = 5 animals were used to create these graphs. Paired t-test was used to produce p-values. p-values smaller than 0.05 are considered to be significant and marked with '*.' Values between 0.05 and 0.1 are shown as p<0.1, where relevant, to highlight a trend in the data.

(AngII infusion), and vasodilated (ACh infusion) conditions. Our data confirm that blood flow in renal microcirculation tends to demonstrate clustered, frequency-locked activity, with the clustering size and tendencies changing depending on the animal condition. Synchronization tends to be stronger, and cluster size is larger during AngII infusion, with synchronization degree $\overline{S} = 0.54 \pm 0.09$ and average synchronization duration ranging from ≈40 ± 13 to $24 \pm 6\%$ of the observed time depending on the distance between vessels. During the ACh infusion, on the contrary, synchronization seems to be disrupted, with ($\overline{S} = 0.36 \pm 0.1$) and average synchronization duration $\approx 15 \pm 7 - 13 \pm 7\%$ of the observed time. Finally, in the normotensive condition, we observed mixed behavior with highly variable synchronization degree $\overline{S} = 0.45 \pm 0.22$ and duration ranging from ≈30 ± 27 to $21 \pm 21\%$. A possible explanation for the stronger synchronization during the AngII infusion will be a stronger hemodynamic coupling due to increased vascular resistance. It is also supported by the increased number of vessels synchronized in anti-phase during the infusion, which, as we predicted using mathematical modeling (*Postnov et al., 2012*), is a natural consequence of stronger hemodynamic coupling.

We also observed phase waves that travel over the TGF-frequency clustered vessels (*Figures 4 and 5B*). Interestingly, the direction of the phase waves appears to be not predetermined by the vascular structure - depending on the flow dynamics and other, yet unknown factors, it can change even within the same animal (an example is shown in *Figure 4*). Such behavior might be related to diffusive interaction between nephrons or the formation of synchronization centers (pacemakers) at different locations. The presence of phase waves supports the deterministic nature of synchronization rather than random entrainment at the same dominant TGF frequency. Similar effects are well studied in excitable media with diffusion interaction mechanisms such as brain (*Dahlem et al., 2010*; *Shibata and Bures, 1972*) and heart (*Alonso et al., 2016*) tissues. Such similarity might suggest the presence of a diffusive mechanism in the inter-nephron interaction or a long-distance fast electrical signaling, e.g., via conducted vasoreactivity. However, a detailed exploration of the topic would require additional experiments combining multi-scale blood flow and structural imaging methods.

In our experiments, we imaged 1.5 × 1.5 mm² field of view in five animals with $94.8 \pm 15.66$ individual segmented vessels on average, each of which is 1o–30 µm in diameter. While it provides an estimate of synchronization in the nephrons activity, direct translation from individual segmented vessels to nephrons is challenging and requires further exploration. Factors that are critical to consider are: (i) penetration depth of LSCI when applied to renal imaging, (ii) type of the vessels in the field of view, and (iii) topology of the nephro-vascular network. While in theory, LSCI can collect the blood flow signal from as deep as 300–400 µm, in practice, visually resolvable signal typically comes from top 50 to 150 µm of the vascular structure (*Davis et al., 2014*; *Zheng et al., 2021*). Considering high vascular density close to the renal surface, it would mean that LSCI is likely limited to imaging vessels originating from ≈10000 nephrons in outer 30% of rat renal cortex (*Letts et al., 2017*), which would result in ≈40 nephrons in the 1.5 × 1.5 µm field of view. The larger number of segmented vessels can be explained by their mixed type - afferent and efferent arterioles as well as venules are likely to be segmented. Distinguishing vessel types in LSCI images will require further exploration and registration with high-resolution structural imaging. It, however, does not mean that the observed clusters were limited to, at the most, 40 nephrons. When considering renal vascular topology, it is to be expected that within the 1.5 × 1.5 µm field of view, we observe arterioles that arise from different non-terminal arteries (*Marsh et al., 2017*). Depending on the branching order, each of such arteries can branch into 10 to several hundreds of nephrons, but only a small number of these nephrons will have arterioles reaching close enough to the surface to be segmented from LSCI images. Thus, when a synchronous cluster is observed with LSCI, it is likely to extend several branching orders in depth and reach the size of hundreds and even thousands of nephrons.

While the exact role of inter-nephron communication, co-operative dynamics, and synchronization in kidney-related pathology development is still unclear and requires further exploration, it is evidently altered by the blood pressure and vascular tone. Strong local coupling and in-phase synchronization, while being not evident at the renal artery level (*Postnov et al., 2015a*), are likely to increase pressure variation at the level of afferent arterioles (*Postnov et al., 2012*; *Postnov et al., 2016a*), thus increasing chances of local damage and aggravating pathological condition.

## Materials and methods
### Animal preparation

All experimental protocols were approved by the Danish National Animal Experiments Inspectorate (License 2015-15-0201-00463) and were conducted according to the American Physiological Society guidelines. Male Sprague Dawley rats (Taconic, Denmark) with average weight ≈290 g (n = 5) were used. Before starting surgical procedures, animals were anesthetized in a chamber with 8% sevoflurane. During the surgery, sevoflurane concentration was reduced to a final concentration of ≈2%. Two catheters were inserted in the right jugular vein to allow continuous systemic infusion of drugs and saline. Another catheter was inserted in the carotid artery to measure mean arterial pressure with a pressure transducer (Statham P23-dB, Gould, Oxnard, CA). Then tracheotomy was performed, after which the rat was placed on a servo-controlled heating table maintaining body temperature at 37°C and connected to a mechanical animal ventilator (60 breaths/min; 8 ml/kg bodyweight). To avoid secondary heartbeat and breathing artifacts, Nimbex (muscle relaxant, Sigma) was administered in a concentration of 0.85 mg/ml, first as a bolus injection of 0.5 ml, followed by a continuous intravenous infusion at a rate of 20 µl/min. The left kidney was then exposed, and the left ureter was catheterized to ensure free urine flow. To reduce motion artifacts and avoid drying the kidney surface during the experiment, we placed the kidney in a plastic fixation holder, covered it with warm agarose solution (1% Agarose, Sigma, 99% saline), and put a thin (0.1 mm) cover glass on top of the kidney. Metal thread (40 µm in diameter) was bent in a 'U' shape and positioned on top of the cover glass at the flattest location of the kidney surface, marking the region of interest for imaging procedures. Following the surgical procedures, the animal was left to stabilize for 20 min. Experiments were continued only if the mean arterial pressure remained within 100–120 mm Hg during the control period. At the end of the experiment, animals were euthanized by overdose of sevoflurane, followed by cervical dislocation.

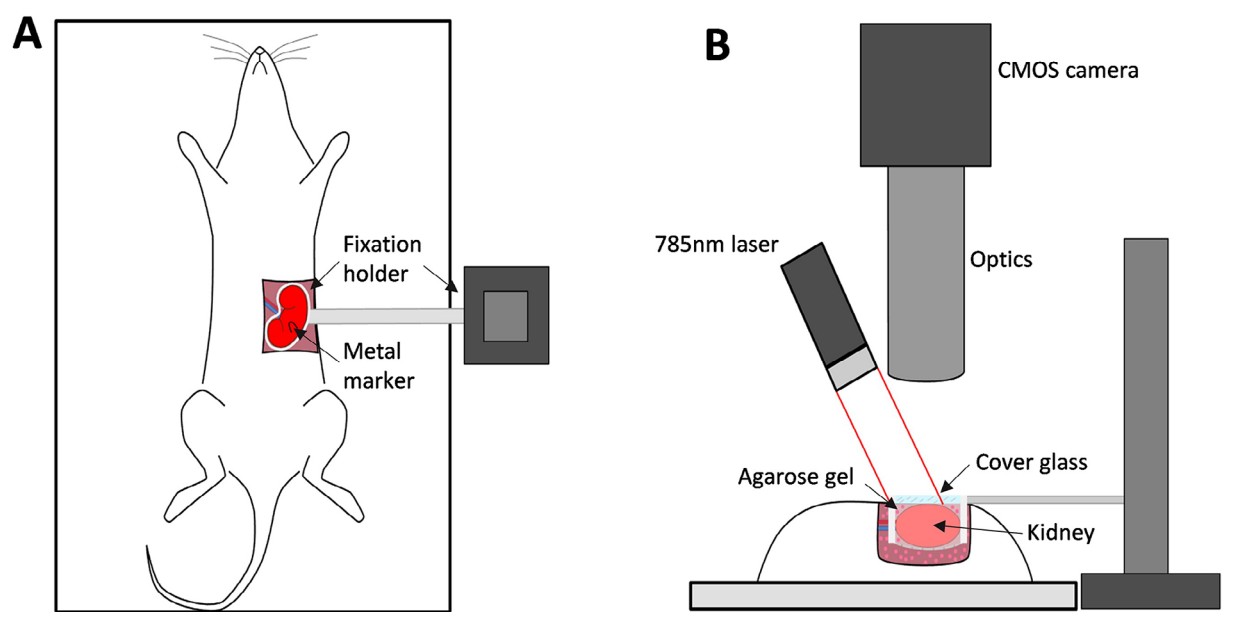

**Figure 6.** The imaging setup and its key components. (**A**) and (**B**) - top and side views respectively.

## LSCI data acquisition

Schematics of the imaging setup are shown in *Figure 6A, B*. To assess microcirculation in the kidney cortex, we built a high resolution LSCI setup. A single-mode fiber-coupled laser diode (785 nm, LP785-SF100, Thorlabs, USA) controlled with a laser driver and temperature controller (CLD1011LP, Thorlabs, USA) was used to deliver coherent light onto kidney surface with a power density of approximately 10 mW/cm$^2$, providing an optimal signal-to-noise ratio (*Postnov et al., 2015b*). Backscattered light was collected by a zoom imaging lens (VZM 1000i, Edmund Optics) at ×5 magnification and recorded with a CMOS camera (Basler acA2000-165umNIR, 2048 × 1088 pixels, 5.5 µm pixel size). Exposure time was set to 5 ms, which provides optimal sensitivity to slow flows, such as in capillary perfused tissue (*Yuan et al., 2005*). A subset of 1024 × 1024 pixels was used for recording, from a 1.5 × 1.5 mm region of the renal surface. In addition, a linear polarizing filter was placed in front of the objective to reduce artifacts from reflected light. While the imaging setup in this configuration allows imaging resolution 1 of ≈0.8 × 0.8 µm$^2$/px at the frame-rate of 160 frames per second (fps), in our experiments, we found it optimal, in terms of the field-of-view and data storage, to acquire images at 50 fps and resolution of ≈1.5 × 1.5 µm$^2$/px. The system provides an image resolution of 2.25 µm$^2$/px compared with previously published studies on renal synchronization, which used the system with the image resolution of 45–400 µm$^2$/px (*Holstein-Rathlou et al., 2011*; *Scully et al., 2013*; *Mitrou et al., 2015*). Similarly, the system has a higher temporal resolution (50 fps instead of 25) and utilizes a better light source (*Postnov et al., 2015b*). Combined and considering noise reduction achieved by spatial and temporal averaging, these factors improve the expected signal-to-noise ratio per area unit by at least 30 times. The examples of data collected with the previously used system and the system presented in this study can be seen in *Figure 1—figure supplement 1*. Further information on the role of high-resolution imaging for identifying vessel-specific oscillations frequency and phase can be found in our recent study (*Lee et al., 2022*).

Imaging was performed in a non-stimulated state (control) and following administration of AngII and ACh, respectively. After collecting 20 min (2400 frames) of the baseline data, we initiated continuous administration of AngII at a concentration of 4 ng/ml and an infusion rate of 20 µl/min to cause systemic vasoconstriction. The infusion lasted for 30 min, out of which the first 10 min were allocated for blood flow to stabilize, and the following 20 min were recorded. 15 min after completion of the AngII infusion, we infused ACh at a concentration of 0.0375 mg/ml and a rate of 20 µl/min, first permitting 10 min for blood flow to stabilize, and then recorded subsequent 20 min. Across all experiments, average arterial pressure was 112 ± 2, 127 ± 6, and 100 ± 4 during the control, vasoconstrited

(AngII), and vasodilated (ACh) conditions, respectively. Averaged across all vessels blood flow index has decreased by 17.3±2.3% in the vasoconstricted condition and increased by 27.7±13.3% in the vasodilated condition. For the details on blood flow index measurements across the conditions see *Figure 1—figure supplement 1*.

## Data analysis

### Image registration

To allow high-resolution LSCI of the renal microcirculation, we needed to reduce motion artifacts, as any lateral motion larger than 5–10 µm will prevent accurate estimation of the blood flow and further segmentation of microcirculatory vessels. Unlike brain imaging, when working with the kidney, there is no bone tissue that can be fixed to reduce respiratory motion, and applying even slight pressure on the kidney might result in abnormal blood flow due to the occlusion of the small vessels on its surface. At the same time, raw laser speckle images, or contrast images without temporal averaging, are not suitable for automated registration due to the absence of clear intensity landmarks (*Postnov et al., 2016b*). To resolve this issue, we placed a 'U' shaped metal marker on the cover glass, which moves along with the kidney, as described above. As the first step of analyzing the data, the marker is segmented in all frames via thresholding and then used to estimate the translation type geometrical transformation required to register images. Estimated geometrical transformation is then applied to the raw laser speckle images prior to performing the contrast analysis.

### Contrast analysis

Registered laser speckle images were processed to calculate temporal contrast $K = \frac{\sigma(I)}{<I>}$, where $\sigma(I)$ and $< I >$ are SD and mean of pixel intensity over 25 frames (*Postnov et al., 2016b*). Contrast values were then converted to the blood flow index as $BFI = 1/K^2$, which are then used in the ensuing analysis.

### Vessels segmentation

To segment individual microcirculatory vessels, we calculated averaged in time blood flow index (BFI) images and applied adaptive thresholding (MATLAB) to them. Automated segmentation was followed by manual clean-up, where we removed artifacts and occasional large surface vessels. We then calculated blood flow dynamics for each segmented individual microcirculatory vessel by averaging BFI values in pixels belonging to this vessel.

### Synchronization analysis

To study synchronization patterns between microcirculatory vessels and, thus, obtain insight into inter-nephron communication, we apply continuous wavelet transform analysis (Morse wavelet, MATLAB) to segmented vessels' BFI. We identified the frequency and phase of dominant periodic activity in the 0.015–0.05 Hz frequency band associated with the TGF mechanism. Vessels with less than 10% prominence of the activity peak were discarded and not used for synchronization analysis. In this study, we consider blood flow in different segmented vessels to be synchronized whenever their dominant frequencies match. To quantify blood flow synchronization over the field of view, we analyzed phase differences between synchronized vessels, average synchronization duration, and its dependency on the distance between vessels, and the probability of the vessel's blood flow to be synchronized with N% of the vessels in the field of view. To provide a 'single-value' characterization of the synchronization at a given moment of time, we also introduced the synchronization degree parameter $S$:

$$S(t) = \sqrt{L(t)/(N * (N - 1))}, \tag{1}$$

where $L$ is a number of frequency matching pairs of vessels, $N$ is a total number of observed segmented vessels, and $t$ is the time. $S$ represents the relation of the observed number of frequency matching pairs to their maximum possible number. Thus $S = 1$ corresponds to all segmented blood vessels having the same dominant frequency, while $S = 0$ to all segmented blood vessels having a different dominant frequencies. However, it is important to notice that $S = 0$ is impossible to reach due to the discrete nature of the measured data and the analysis. In our case, the 0.015–0.05 Hz range is split into 18 fixed values, so that if there were more than 18 vessels, it became unavoidable for some of them to have an identical dominant frequencies. In practice, for 100 vessels with randomly chosen

dominant frequencies, the minimum observed $S$ would be ≈0.25±0.06, which can be confirmed with a simple computational experiment.

## Statistical analysis

Paired t-test was applied to compare results between control, AngII infusion, and ACh infusion. p-values greater than 0.05 are reported as not significant. Results were expressed as mean ± SD unless indicated otherwise.

# Additional information

### Funding

| Funder | Grant reference number | Author |
| --- | --- | --- |
| Novo Nordisk Fonden | | Dmitry Postnov |
| Lundbeckfonden | | Dmitry Postnov |

The funders had no role in study design, data collection and interpretation, or the decision to submit the work for publication.

### Author contributions

Dmitry Postnov, Conceptualization, Data curation, Formal analysis, Funding acquisition, Investigation, Methodology, Project administration, Resources, Software, Supervision, Validation, Visualization, Writing - original draft, Writing - review and editing; Donald J Marsh, Will A Cupples, Niels-Henrik Holstein-Rathlou, Olga Sosnovtseva, Conceptualization, Methodology, Writing - original draft, Writing - review and editing

### Author ORCIDs

Dmitry Postnov http://orcid.org/0000-0002-9708-8453

### Ethics

All experimental protocols were approved by the Danish National Animal Experiments Inspectorate (License 2015-15-0201-00463) and were conducted according to the American Physiological Society guidelines.

### Decision letter and Author response

Decision letter https://doi.org/10.7554/eLife.75284.sa1
Author response https://doi.org/10.7554/eLife.75284.sa2

# Additional files

### Supplementary files
• Transparent reporting form

### Data availability

The data underlying this article are available at public data repository (Dryad): https://doi.org/10.5061/dryad.g79cnp5r2.

The following dataset was generated:

| Author(s) | Year | Dataset title | Dataset URL | Database and Identifier |
| --- | --- | --- | --- | --- |
| Postnov D, Marsh D, Cupples W, Holstein-Rahtlou N, Sosnovtseva O | 2021 | Data from: Synchronization in renal microcirculation unveiled with high-resolution blood flow imaging | https://dx.doi.org/10.5061/dryad.g79cnp5r2 | Dryad Digital Repository, 10.5061/dryad.g79cnp5r2 |

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
