## [Editor Report]

The new technical advance reported here, high-resolution laser speckle contrast imaging of many microvessels simultaneously and application to the entire rat kidney surface should be of great interest to readers interested in general vascular physiology and especially renal hemodynamics research. The observed large vessel cluster synchronization and phase wave formation phenomena will help further the understanding of renal blood flow regulation in health and disease.

---

## [Decision Letter]

**Decision letter after peer review:**

Thank you for submitting your work entitled "Synchronization in renal microcirculation unveiled with high-resolution blood flow imaging" for further consideration by *eLife*. Your article has been reviewed by 2 peer reviewers, and the evaluation has been overseen by a Reviewing Editor and overseen by Martin Pollak as Senior Editor. The reviewers have opted to remain anonymous.

There are some issues that need to be addressed. In particular, we would like to see some evidence regarding how this method compares with previous methodologies (see Reviewer 1). We would also like to see the other specific points raised by the reviewers addressed to our satisfaction.

*Reviewer #1 (Recommendations for the authors):*

The key observations of the study are the synchronization of oscillations in nephron blood flow oscillation frequency at frequencies previously demonstrated to be governed by tubuloglomerular feedback (TGF). Using laser speckle microscopy, the authors observe similar frequency of blood flow oscillation across many nephrons, spanning an area over 1 mm2. The authors then characterize this synchronization of oscillation frequency in response to systemic administration of angiotensin II and acetylcholine to induce systemic vasoconstriction and vasodilation and observe that angiotensin II increases differences in phase, while acetylcholine decreases the duration of synchronization. The authors claim that this manuscript presents a novel laser speckle microscopy system to enable characterization of blood flow across a larger area of the kidney, yet the system is not characterized for resolution or field of analysis nor is it quantitatively compared to previous systems. Therefore, the merits of the manuscript rest on the characterization of nephron synchronization, which is largely correlative and lacking in data presentation and controls.

– In the abstract, the authors claim that this manuscript introduces "an approach for high-resolution laser speckle imaging." As motivation for this work, in the introduction, the authors state, "The methods lacked resolution, both spatial and temporal, as well as signal-to-noise ratio, to confirm it convincingly. In this paper we further advance renal blood flow imaging methodology…" However, the authors do not characterize their new system, in terms of resolution (spatial and temporal) and signal-to-noise ratio. Therefore, the advances claimed cannot be evaluated against previous methods.

– In Figure 2, the length scale of subpanels A-C is not indicated. It's mentioned in the discussion to be 1.5 x 1.5 mm, but it is not indicated on the figure or in the caption. How are the size of the dots determined? Also, given the thresholding and image processing required to generate this data set, it is difficult to understand how the dots presented in the figure relate back to the structure of the kidney. Raw data and images with the nephrons indicated (as in Figure 1) should be provided.

– Similarly, it is difficult to understand how representative the data provided are. While the analysis has been performed on multiple rats (as indicated in the data of Figure 3), there is no information on the variation from measurement to measurement vs. animal to animal, regarding segmentation, synchronization, or effects of administered compound.

– In all conditions, the authors administer the vasoconstriction agent followed by vasodilation. Does this ordering affect the synchronization?

– In the discussion, the authors claim that TGF-frequency clustered waves propagate in a manner where "the direction of the phase is not predetermined by the vascular structure – depending on the flow dynamics and other, yet unknown factors…" What data justify the claim regarding structure and flow dynamics?

*Reviewer #2 (Recommendations for the authors):*

In this study, Postnov et al., are reporting the application of a new technical advance, further improvement of their recently developed imaging technology using LSCI. In the present work, they performed high-resolution LSCI imaging measurement and analysis of renal blood flow dynamics in multiple individual small renal vessels simultaneously in the entire surface of intact rat kidneys. They found synchronized clusters of blood vessels spanning multiple (>40) nephrons, increased or decreased level of synchronization in response to AngII or Ach, respectively, and the development of space-dependent phase waves. From these results the authors concluded that strengthened hemodynamic coupling due to increased vascular tone, alternating synchronization (pacemaker) centers likely played important roles in the observed phenomena.

Strengths:

The paper is well-written, and the study appears to be well performed. The results are clearly presented and support the conclusions. The presently applied imaging technical advance will be helpful to better understand the detailed, nephron-to-nephron level functional features and role of renal hemodynamics in physiological and disease conditions. However, there are a few points that could be more clarified.

Weaknesses:

1. It would be more convincing if vasoconstricted (Ang II infusion) and vasodilated conditions (Ach infusion) would be validated experimentally by quantitative vessel diameter measurements rather than simply assumed based on the expected vasoactivity of the applied drugs. This is especially important due to the rather low doses applied.

2. Since the LSCI signal strength depends on healthy blood flow rates, vasoconstrictor states may limit the sensitivity of this technique to detect the detailed, single vessel hemodynamic oscillations and the large cluster synchronization features described here. It would be interesting to know the level of total renal blood flow alterations in the presently applied vasoactive treatment conditions.

3. As it is understandable for a technical advance type of report, the study is rather descriptive, and the study design is lacking mechanistic examination of the observed synchronization and phase waves phenomena. Although the presence of fast electric or diffusive coupling that generates long-distance nephron-to-nephron communication was discussed. The applied vasoactive compounds, AngII and Ach have numerous extra-renal effects, for example on nerve activity which may be involved in cluster synchronization. It would be interesting to examine how the large cluster synchronization and phase waves are altered in kidneys one week after denervation. Since a vasoconstrictor state was associated with strengthened synchronization, one is wondering if stronger synchronization could be a sign of increased sympathetic nerve activity.

4. The physiological significance of nephron cluster synchronizations should be clearly stated.

In addition to the many strengths detailed above, I have a few suggestions for further improvement.

1. Vasoconstricted (Ang II infusion) and vasodilated conditions (Ach infusion) need to be validated experimentally by quantitative vessel diameter measurements. Vessel dimensions could be measured using the LSCI images or other higher resolution imaging modalities that could be simultaneously applied.

2. It would be important to show if in the AngII condition the reduced blood flow and reduced sensitivity of the technique could contribute, at least in part, to the observed large clustering. Perhaps including an original, continuous tracing of RBF oscillations before and after AngII treatment would alleviate this concern.

3. Renal denervation could address mechanistic details of the observed large clustering and phase wave phenomena.

4. One or two additional sentences in the introduction should state the physiological significance of nephron synchronization.

---

## [Author Response]

Reviewer #1 (Recommendations for the authors):The key observations of the study are the synchronization of oscillations in nephron blood flow oscillation frequency at frequencies previously demonstrated to be governed by tubuloglomerular feedback (TGF). Using laser speckle microscopy, the authors observe similar frequency of blood flow oscillation across many nephrons, spanning an area over 1 mm2. The authors then characterize this synchronization of oscillation frequency in response to systemic administration of angiotensin II and acetylcholine to induce systemic vasoconstriction and vasodilation and observe that angiotensin II increases differences in phase, while acetylcholine decreases the duration of synchronization. The authors claim that this manuscript presents a novel laser speckle microscopy system to enable characterization of blood flow across a larger area of the kidney, yet the system is not characterized for resolution or field of analysis nor is it quantitatively compared to previous systems. Therefore, the merits of the manuscript rest on the characterization of nephron synchronization, which is largely correlative and lacking in data presentation and controls.– In the abstract, the authors claim that this manuscript introduces "an approach for high-resolution laser speckle imaging." As motivation for this work, in the introduction, the authors state, "The methods lacked resolution, both spatial and temporal, as well as signal-to-noise ratio, to confirm it convincingly. In this paper we further advance renal blood flow imaging methodology…" However, the authors do not characterize their new system, in terms of resolution (spatial and temporal) and signal-to-noise ratio. Therefore, the advances claimed cannot be evaluated against previous methods.

We have added clarification in the text and a supplementary figure (Figure 1—figure supplement 1) that compares measurements taken with our system versus measurements taken with the system used for the renal LSCI before. Further information on the importance of high-resolution imaging and vessel masking approach can be found in our recent study (see Lee at all 2022).

Changes:

Text added (Lines 253-263): “The system provides an image resolution of 2.25µ^2^ compared to previously published studies on renal synchronization, which used the system with the image resolution of 45 – 400µ^2^ {holstein2011nephron,scully2013detecting,mitrou2015laser}. Similarly, the system has a higher temporal resolution (50 frames per second instead of 25) and utilizes a better light source {postnov2015improved}. Combined and considering noise reduction achieved by spatial and temporal averaging, these factors improve the expected signal to noise ratio per area unit by at least 30 times. The examples of data collected with the previously used system and the system presented in this study can be seen in Supplementary Figure 1. Further information on the role of high-resolution imaging for identifying vessel-specific oscillations frequency and phase can be found in our recent study{lee2022multi}.”

– In Figure 2, the length scale of subpanels A-C is not indicated. It's mentioned in the discussion to be 1.5 x 1.5 mm, but it is not indicated on the figure or in the caption. How are the size of the dots determined? Also, given the thresholding and image processing required to generate this data set, it is difficult to understand how the dots presented in the figure relate back to the structure of the kidney. Raw data and images with the nephrons indicated (as in Figure 1) should be provided.– Similarly, it is difficult to understand how representative the data provided are. While the analysis has been performed on multiple rats (as indicated in the data of Figure 3), there is no information on the variation from measurement to measurement vs. animal to animal, regarding segmentation, synchronization, or effects of administered compound.

Colour-coded circles in Figure 2 are a way to provide clear visualization of the dynamics identified in separate vessels. The size of the circles is identical and holds no information, while the location is defined by the centre of mass of corresponding vessels. All the panels presented in Figures 2 and 4 serve to demonstrate examples of observed dynamics and correspond to different recordings from the same animal shown in Figure 1 A. Thus, we already show the data indicating the positions as asked by the reviewer.

The information on variation between animals is provided in Figures 3 and 5 and is discussed in detail in the text. The average and deviation in figures 3 and 5 are calculated over all 5 animals.

We have clarified the text and scale bars to all relevant figures.

Changes:

Scale bars added to figures 2 and 4.

Text added (Lines 95-100): “To simplify the visualisation of frequencies and phases, we generate maps where coloured circles represent the segmented vessels (outlined in magenta in Figure~{Figure 1}(A)), with the circle centre located in the centre of the mass of the vessel. The circle's colour represents the peak TGF frequency observed in the corresponding vessel, while the size of the circles is identical and does not hold any information.”

– In all conditions, the authors administer the vasoconstriction agent followed by vasodilation. Does this ordering affect the synchronization?

We should highlight that the study aims not to see the effect of ACh after infusion of AngII, but to observe synchronization in three different conditions (normal, vasoconstricted, vasodilated). The infusions order protocol was designed for this purpose considering the washout time of the drugs. AngII action is expected to be entirely gone (and receptors recover) within 25 minutes of the resting time after the infusion is over. Changing the order should not change the result as long as the washout times taken into account.

– In the discussion, the authors claim that TGF-frequency clustered waves propagate in a manner where "the direction of the phase is not predetermined by the vascular structure – depending on the flow dynamics and other, yet unknown factors…" What data justify the claim regarding structure and flow dynamics?

It is justified by the direction of the phase waves changing in the same animal depending on time and condition (meaning that it changes within the same vascular structure). Figure 2 shows an example: in the top panels, the phase wave travels from bottom-left to top-right, while in the bottom panels, it travels from top left to bottom right. As we write in the discussion, however, confirming and understanding the mechanism of the phase waves requires further analysis and experiments that would combine multi-scale blood flow and structural imaging.

Changes:

Text changed (Line 177): "is not predetermined" to "appears to be not predetermined".

Text added (Lines 186-187): "However, a detailed exploration of the topic would require additional experiments combining multi-scale blood flow and structural imaging methods."

Reviewer #2 (Recommendations for the authors):In this study, Postnov et al., are reporting the application of a new technical advance, further improvement of their recently developed imaging technology using LSCI. In the present work, they performed high-resolution LSCI imaging measurement and analysis of renal blood flow dynamics in multiple individual small renal vessels simultaneously in the entire surface of intact rat kidneys. They found synchronized clusters of blood vessels spanning multiple (>40) nephrons, increased or decreased level of synchronization in response to AngII or Ach, respectively, and the development of space-dependent phase waves. From these results the authors concluded that strengthened hemodynamic coupling due to increased vascular tone, alternating synchronization (pacemaker) centers likely played important roles in the observed phenomena.Strengths:The paper is well-written, and the study appears to be well performed. The results are clearly presented and support the conclusions. The presently applied imaging technical advance will be helpful to better understand the detailed, nephron-to-nephron level functional features and role of renal hemodynamics in physiological and disease conditions. However, there are a few points that could be more clarified.Weaknesses:1. It would be more convincing if vasoconstricted (Ang II infusion) and vasodilated conditions (Ach infusion) would be validated experimentally by quantitative vessel diameter measurements rather than simply assumed based on the expected vasoactivity of the applied drugs. This is especially important due to the rather low doses applied.

We do confirm the effect of the vasoactive drugs by measuring average arterial blood pressure via carotid artery catheter. As it is stated at the end of the Methods section 4.2 the average arterial pressure was observed to be 112±2 in control, 127±6 during vasoconstriction and 100±4 during vasodilation which agrees well with our expectations. The effect is further confirmed by changes in blood flow index measured with LSCI.

Regarding the diameter measurements – reliable measurements of star-vessels diameter with LSCI do not appear to be feasible due to specifics of renal vascular geometry or unless the field of view is further reduced which will make synchronization studies impossible. It could be done if LSCI is combined with multi-photon microscopy, which is out of the scope for the present paper but is part of studies that we plan in future.

2. Since the LSCI signal strength depends on healthy blood flow rates, vasoconstrictor states may limit the sensitivity of this technique to detect the detailed, single vessel hemodynamic oscillations and the large cluster synchronization features described here. It would be interesting to know the level of total renal blood flow alterations in the presently applied vasoactive treatment conditions.

We thank reviewer for a suggestion and have added supplementary figure S2, which shows average blood flow index in all three conditions with and without normalization. From it you can see that blood flow index is reduced by 20% during AngII infusion and increased by 30% during Ach infusion.

We should mention, however that while LSCI sensitivity depends on the relation between particles speed and the exposure time, the effect can be neglected in our study. We use exposure time = 5ms, which is optimal for slow flows (e.g. capillary perfused tissue) and matches our application well. Even with vasoactive agents induced the blood flow index change is +-30%, while noticeable loss of sensitivity is expected for flows which are much faster or slower (e.g. 10 times) (see Yuan et al., 2005 [38])

Changes:

Text added (Lines 247-248): “Exposure time was set to 5 ms, which provides optimal sensitivity to slow flows, such as in capillary perfused tissue{yuan2005determination}.”

Text added (Lines 273-276): “Averaged across all vessels blood flow index has decreased by 17.3±2.3% in the vasoconstricted condition and increased by 27.7±13.3% in the vasodilated condition. For the details on blood flow index measurements across the conditions see Supplementary Figure 2.”

Figure added: Figure 1 —figure supplement 2.

3. As it is understandable for a technical advance type of report, the study is rather descriptive, and the study design is lacking mechanistic examination of the observed synchronization and phase waves phenomena. Although the presence of fast electric or diffusive coupling that generates long-distance nephron-to-nephron communication was discussed. The applied vasoactive compounds, AngII and Ach have numerous extra-renal effects, for example on nerve activity which may be involved in cluster synchronization. It would be interesting to examine how the large cluster synchronization and phase waves are altered in kidneys one week after denervation. Since a vasoconstrictor state was associated with strengthened synchronization, one is wondering if stronger synchronization could be a sign of increased sympathetic nerve activity.

We are thankful to reviewer for the suggestion. It is indeed crucial to study synchronization under other conditions including the denervation. While it is out of the scope in the present study, we do aim to cover it in the future studies.

4. The physiological significance of nephron cluster synchronizations should be clearly stated.

We have clarified it in the text. Although we should bring it to the attention that since the topic is not well explored such discussion has a theoretical and speculative nature, requiring multiple studies to be done to come to specific conclusions.

Changes:

Text added (Lines 53-62): “Adaptive synchronization across the renal microvascular network would increase the efficiency and dynamic range of autoregulation by engaging more pre-glomerular resistance

{marsh2019nephron,zehra2021tubuloglomerular}, preventing transmission of high systemic pressure to the glomeruli, which could lead to progressive glomerular and vascular injury. In addition, long-distance synchronization would argue strongly that renal autoregulation is a distributed process that can ensure an optimized oxygenation-perfusion matching and adjust to various internal or environmental conditions {sosnovtseva2007synchronization, mitrou2015laser}. On the other hand, large scale in-phase synchronization might become a mechanism leading to glomerular injury as it would substantially increase local pressure variation {postnov2012dynamics}.”

In addition to the many strengths detailed above, I have a few suggestions for further improvement.1. Vasoconstricted (Ang II infusion) and vasodilated conditions (Ach infusion) need to be validated experimentally by quantitative vessel diameter measurements. Vessel dimensions could be measured using the LSCI images or other higher resolution imaging modalities that could be simultaneously applied.

Please see Reply 1 above

2. It would be important to show if in the AngII condition the reduced blood flow and reduced sensitivity of the technique could contribute, at least in part, to the observed large clustering. Perhaps including an original, continuous tracing of RBF oscillations before and after AngII treatment would alleviate this concern.

Please see Reply 2 above

3. Renal denervation could address mechanistic details of the observed large clustering and phase wave phenomena.

Please see Reply 3 above

4. One or two additional sentences in the introduction should state the physiological significance of nephron synchronization.

Please see Reply 4 above